
# Data limitations and potential of hourly and daily rainfall thresholds for shallow landslides

Elena Leonarduzzi[1,2] and Peter Molnar[1]

[1]Institute of Environmental Engineering, ETH Zurich, Switzerland
[2]Swiss Federal Institute for Forest, Snow and Landscape Research WSL, Birmensdorf, Switzerland

**Correspondence:** Elena Leonarduzzi (leonarduzzi@ifu.baug.ethz.ch)

**Abstract.** Rainfall thresholds are a simple and widely used method to predict landslide occurrence. In this paper we provide a comprehensive data-driven assessment of the effects of rainfall temporal resolution (hourly versus daily) on landslide prediction performance in Switzerland, with sensitivity to two other important aspects which appear in many landslide studies–the normalisation of rainfall, which accounts for local climatology, and the inclusion of antecedent rainfall as a proxy of soil water

state prior to landsliding. We use an extensive landslide inventory with over 3800 events and several daily and hourly, station and gridded rainfall datasets, to explore different scenarios of rainfall threshold estimation. Our results show that although hourly rainfall did show best predictive performance for landslides, daily data were not far behind, and the benefits of hourly resolutions can be masked by the higher uncertainties in threshold estimation connected to using short records. We tested the impact of several typical actions of users, like assigning the nearest raingauge to a landslide location and filling in unknown

timing, and report their effects on predictive performance. We find that localisation of rainfall thresholds through normalisation compensates for the spatial heterogeneity in rainfall regimes and landslide erosion process rates and is a good alternative to regionalisation. On top of normalisation by mean annual precipitation or a high rainfall quantile, we recommend that non-triggering rainfall be included in rainfall threshold estimation if possible. Finally, we demonstrate that there is predictive skill in antecedent rain as a proxy of soil wetness state, despite the large heterogeneity of the study domain, although it may not be

straightforward to build this into rainfall threshold curves.

## 1 Introduction

Landslides are a natural hazard that affects alpine regions worldwide resulting in substantial economic losses and human casualties (Kjekstad and Highland, 2009). Landslides can be initiated by different triggering factors, mainly rainfall and earthquakes. Economic losses connected to landsliding are estimated to be between 0.5 and 5 Billion USD annually for the European Alps

regions (e.g., Salvati et al., 2010; Trezzini et al., 2013; Klose, 2015; Kjekstad and Highland, 2009), and similar losses are also reported for Canada and the United States (e.g., Kjekstad and Highland, 2009; Schuster, 1996). Petley (2012) carried out a global study over a 7 year period (2004-2010) and found a total of 2620 nonseismically triggered landslides causing 32'322 fatalities. Clearly, the socio-economic impact of landslides is large and this natural hazard requires attention in the form of risk mapping, better prediction, and early warning systems.



The focus in this work is on rainfall-induced shallow landslides, which are the predominant type of landslides in Switzerland and other alpine environments. These are landslides where the entire soil (upper regolith) fails along a weathered bedrock interface, and they develop quickly leading to mass failure following soil-saturating rainfall (e.g., Highland et al., 2008). Despite their smaller size, these landslides can be widespread and have the potential to damage infrastructure (railways, roads), homes, and even lead to fatalities. For instance in Switzerland, a total of 520 Million Euros in damage was recorded in the

period 1972–2007 and 32 people lost their life due to shallow landslides (Hilker et al., 2009).

    One of the most widespread approaches for the prediction of triggering conditions leading to rainfall-induced landslides is that of rainfall thresholds (e.g., Stevenson, 1977; Caine, 1980; Guzzetti et al., 2007), which are used operationally in many countries (e.g., see reviews in Guzzetti et al., 2019; Piciullo et al., 2018). These can be based on any rainfall property, but most frequently are assumed to be power law curves in the intensity-duration (ID) or the total rainfall-duration (ED) space.

The reasoning behind this choice is that two different storm types may be responsible for the initiation of landslides: short and intense, or long-lasting and typically less intense. Many approaches exist to formulate and estimate ID or ED curves, and they differ in the accuracy of the landslide inventory, the rainfall records used, the definition of rainfall events, the statistical methodology for threshold definition, and the validation technique, among others (see review in Segoni et al., 2018).

    One of the main aspects in which the approaches differ is the choice of rainfall temporal resolution, typically forced by

data availability. The short and intense events responsible for local soil saturation and triggering of landslides are usually associated with convective activity, which can last for just a few hours (e.g., Molnar and Burlando, 2008). For this reason, hourly thresholds are expected to be more appropriate for landslide prediction. There is some evidence for this in the literature. For example, Marra (2019) shows the underestimation of rainfall thresholds as the temporal resolution of rainfall is coarsened with a numerical experiment, while Gariano et al. (2019) demonstrate a similar effect on a real case dataset where reported

landslides are combined with rain gauge records aggregated to different temporal resolutions. At the same time, daily rainfall thresholds or ID curves may also exhibit good predictive power for landsliding, e.g. as shown in a comprehensive analysis in Switzerland (Leonarduzzi et al., 2017). So the question arises, how does the temporal resolution of the rainfall data actually affect landslide prediction?

    We address this question with a data analysis experiment, where we take into account realistic conditions and consequences

of different data resolutions. For example, choosing a higher resolution (e.g. hourly) has several undesirable consequences: (a) rainfall records are typically shorter (hourly records are only available in automatic networks in more recent decades), (b) rainfall records are likely to be less dense in space leading to poorer matching with landslide locations, and (c) landslide inventories are typically less rich (required timing and not only date of occurrence) or more uncertain, especially for older events that were reconstructed from newspaper articles or other indirect sources. All these aspects have to be taken into consideration

in an objective analysis of the effects of temporal resolution on rainfall thresholds.

    In this paper we undertake such an analysis with the high quality landslide database available in Switzerland (Hilker et al., 2009) and several high quality rainfall records available for Switzerland from Meteoswiss. We expand the impact of temporal resolution (hourly versus daily) on landslide prediction with sensitivity to two other important aspects (analysis steps) which appear in many landslide studies: the normalisation of rainfall which accounts for local meteorological properties (e.g. Marc




et al., 2019), and the inclusion of antecedent rainfall which provides additional information on soil state prior to landsliding (Glade et al., 2000; Godt et al., 2006; Mirus et al., 2018). The objectives of the paper therefore are (a) to provide an extensive realistic comparison between hourly and daily rainfall data for the definition of rainfall thresholds, (b) to compare different strategies for the normalisation of rainfall thresholds, and (c) to explore whether antecedent rainfall does provide added predictive power at the regional/national scale.

## 2  Data and Methods

We use several rainfall datasets and a landslide inventory (Hilker et al., 2009) (Section 2.1) to derive objective landslide-triggering rainfall thresholds at the daily and hourly scale using two different statistical methods (TSS maximisation and frequentist approach) (Section 2.2), and to address some of the issues associated with higher temporal resolution data, such as the absence of accurate timing information for landslide occurrence (Section 2.3) and the lower quality (density) of rainfall

data (Section 2.4). We follow up with methods which quantify the impact of rainfall threshold normalisation (Section 2.5) and the added power of antecedent rainfall on landslide prediction (Section 2.6).

### 2.1  Rainfall and landslide data

The rainfall datasets used differ by type of measurement, duration of record, and temporal and spatial resolutions (Figure 1 and Table 1). The daily product (Rainfall Daily Interpolated, RDI) is the longest record (1972–2018), containing daily sums (6

am to 6 am) over 1×1km cells covering Switzerland. It is obtained by interpolating daily measurements from approximately 420 rain gauges, using the climatology (intended here as anomaly relative to the monthly mean precipitation over the reference period 1971–1990), and regionally varying precipitation-topography relationship (procedure explained in details in Frei and Schär, 1998).

The hourly station rainfall dataset (Rainfall Hourly Gauges, RHG) is the collection of the hourly rainfall timeseries measured
continuously since 1981 at 45 gauges across the country (green dots in Figure 1). We used two hourly gridded raninfall datasets which were derived from RDI and are reported at the same spatial resolution 1×1km and the daily sums match that of the corresponding RDI cell. The first dataset (Rainfall Hourly Interpolated Gauges, RHIG) is computed by disaggregating the daily sum RDI into hourly intensities by using the hourly fractions recorded at the nearest hourly gauge (RHG). The second dataset (Rainfall Hourly Interpolated Radar, RHIR), instead uses an hourly composite of radar measurements NASS (Joss et al.,

1998; Germann and Joss, 2004; Germann et al., 2006) for the disaggregation (procedure explained in details in Wüest et al., 2010). Due to the quality of the radar composite, we expect RHIR to be more accurate than RHIG in-between stations. In fact, the hourly gauge network measuring continuously since 1981 is quite sparse and it is likely to miss heavy rainfall intensities especially during convective storms.

The four different rainfall records are combined with the landslides extracted from the Swiss flood and landslide damage
database (Hilker et al., 2009). This databases collects floods, debris flows, landslides, and rockfalls that produced damage in Switzerland since 1972. Of the total reported landslides in the period 1972–2018 we selected those with known location and





date. Then, depending on the rainfall dataset used, the timeframe is modified, and for hourly analysis a further selection is made of the entries with known timing (number of landslides per rainfall dataset reported in Table 1).

## 2.2 Rainfall thresholds

The methodology for the definition of rainfall thresholds follows the statistical procedure introduced in Leonarduzzi et al. (2017). First we separate the rainfall timeseries into events, by considering a minimum amount of dry hours in between. We choose 24 hours for daily rainfall data and 6 hours for hourly rainfall data, after an optimisation within a range of 2-12 hours, which is the amount of dry hours expected to separate individual storms.

Then we classify rainfall events as triggering events if a landslide happens during or immediately after the event, and non-triggering otherwise. We compute the event duration, total rainfall, mean and maximum rainfall intensity for each event. We then define optimal thresholds for each of the precipitation characteristics by finding the threshold that maximises the True Skill Statistic $TSS = specificity + sensitivity - 1$, where sensitivity is the rate of true positives and specificity is the rate of true negatives. Additionally we also define total rainfall $E$ versus duration $D$ (ED) thresholds in the form of a power law function, $E = a \cdot D^b$, by optimising the two parameters $a$ and $b$ through TSS maximisation. As a reference, we provide also the
results for the thresholds defined following the frequentist approach, first introduced in Brunetti et al. (2010), which is one of the most widely used methods for ED fitting (e.g., Peruccacci et al., 2012; Vennari et al., 2014; Gariano et al., 2015; Iadanza et al., 2016; Melillo et al., 2018; Roccati et al., 2018). The optimum threshold in this case is based on triggering events only. The exponent $b$ is obtained by fitting the ED pairs with a line in log-log space. The intercept $a$ is adjusted to match a chosen exceedance probability (in this paper we use the 5% exceedance probability as a reference).

For all analyses based on the gridded rainfall products, we consider the rainfall timeseries for each susceptible cell, for which we define rainfall events following the procedure explained above. Susceptible cells are those rainfall cells in which at least one landslide was recorded in the respective timeframe of each dataset in Table 1.

## 2.3 Inaccurate landslides timing: triggering and peak intensities

One problem we face when utilising hourly rainfall records, is that the actual timing of historical landslides is typically not
available or very uncertain/inaccurate. For instance, Guzzetti et al. (2007) report that out of the 2626 rainfall events associated with shallow slope failures globally, only 26.3% had information about the date of occurrence and only 5.1% also the timing. Although a common approach to compensate for the lack of accurate landslide timing is to assign the landslide to the rainiest hour within a certain time window, the effect of this approximation is not well known. Peres et al. (2018) showed the potential impact of timing and date uncertainty using synthetic databases by coupling a stochastic weather generation and a physically
based hydrological and slope stability model. Staley et al. (2013) showed, using a precise debris-flow database, that using peak rainstorm intensity instead of the actual triggering intensity, results in an overestimation of the ID threshold.

We study the wrong timing effect similarly to Staley et al. (2013) by introducing two scenarios as alternatives to the actual landslide database: one in which we assume that when the day of a landslide is known its timing is assigned to the most intense rainy hour within the day (this is the case of Staley et al., 2013), and a second alternative in which the timing is assigned





to the rainiest hour within a 48h window centred on the actual timing recorded in the database (this is a hypothetical case which considers the fact that we may not have the right date recorded in the landslide database). Once the timing is altered accordingly, the modified landslide databases are used for the definition of ED thresholds following the same procedure as with the original true database. We carry out this exercise utilising landslides with known time of occurrence recorded between May 2003 and December 2010 (timeframe of RHIR).

## 2.4 Rainfall quality: gauge density and interpolation

In most studies where regional rainfall thresholds are defined, landslides in a region are assigned to the closest rain gauge, sometimes taking into consideration not only distance (Finlay et al., 1997; Godt et al., 2006), but also similarities in topography or other aspects important for precipitation (e.g., Aleotti, 2004; Berti et al., 2012; Gariano et al., 2012; Rossi et al., 2012; Melillo et al., 2018; Vennari et al., 2014). Nikolopoulos et al. (2015) showed that decreasing the density of the rain gauge network, the $b$

parameter of the power law ID curve on average decreases, depending on whether the closest rain gauge is considered (nearest neighbour) or simple interpolation methods are used such as inverse distance weighting or ordinary kriging. In the context of comparing the impacts of daily and hourly rainfall resolutions on landslide thresholds, we recognise that gauge density is very important and we construct an experiment to test the effects of gauge density and accuracy of spatial interpolation.

To do this, we define rainfall thresholds with the "closest rain gauge" based on the very sparse station-based hourly rainfall

record RHG and compare it to the spatially-distributed disaggregated dataset RHIG. The comparison shows the effect of improving an hourly record obtained with a very sparse network, by taking advantage of a daily dataset based on a much denser network and an advanced interpolation method in RDI (Frei and Schär, 1998), merged with hourly station data. We propose two versions of the "closest rain gauge" approach used in many studies. First we assign each landslide to the geographically closest rain gauge and then extract rainfall events for each of the gauges which have at least one landslide (maximum 45 rain

gauges) from the RHG dataset. Second we assign to each susceptible rainfall cell (as defined for the gridded rainfall products) the rainfall of the closest rain gauge, and the event definition is carried out for each of these cells (maximum as many cells as the number of landslides) from the RHIG dataset.

## 2.5 Rainfall normalisation

One of the methods suggested to improve the predictive power of regional rainfall thresholds is to localise them. This can be

done through regionalisation, by dividing the area into homogeneous regions and defining a different threshold for each of them (e.g. Peruccacci et al., 2012; Leonarduzzi et al., 2017; Peruccacci et al., 2017), or by normalisation, that is defining thresholds based on the ratio between the precipitation parameters and a local scaling value, considered to be representative of local rainfall characteristics. Typically, the property chosen is the Mean Annual Precipitation MAP (e.g. Dahal and Hasegawa, 2008; Aleotti, 2004; Guzzetti et al., 2007; Leonarduzzi et al., 2017; Peruccacci et al., 2017), the Rainy-Day Normal RND $= \frac{\text{MAP}}{n}$

where $n$ is the number of rainy days in a year (Guidicini and Iwasa, 1977; Wilson and Jayko, 1997; Guzzetti et al., 2007; Postance et al., 2018), or other precipitation characteristics (e.g. anomaly relative to 10 years return period rainfall in Marc et al., 2019).



In this paper we test in addition to the well established MAP and RDN normalisations also quantiles of event properties and of daily/hourly rainfall as scaling parameters. Note that there are fundamental differences between scaling with MAP, RDN or

rainfall quantiles, in that MAP ignores intermittency of rainfall, while RDN and quantiles are computed only from the rainy hours/days of the rainfall dataset.

## 2.6   Antecedent rainfall

The main criticism raised against rainfall thresholds for landsliding in general, is that they only consider recent/event rainfall, without taking into account the soil status prior to it (e.g., Bogaard and Greco, 2018). To include this antecedent soil moisture

state into rainfall thresholds, several ad-hoc approaches have been introduced with varying levels of complexity and data demand. The simplest of these consists in accumulating rainfall over a fixed duration prior to the triggering event rainfall (e.g., Chleborad, 2003; Frattini et al., 2009). In other studies the fixed duration has been modified to account for vanishing memory in rainfall using the Antecedent Precipitation Index (API), which gives less weight to rainfall contributions further back in time (e.g., Crozier et al., 1980; Crozier, 1986), often relating the decay coefficient to the recession curves of storm hydrographs,

as first suggested by Glade et al. (2000). A further development of the API is the so-called Antecedent Wetness Index, which accounts also for other hydrological variables by removing from antecedent rainfall the potential evapotranspiration, and then following the same approach as API (e.g., Godt et al., 2006). Finally, a few studies use estimates of the real antecedent soil wetness which are based on the soil water balance (Ponziani et al., 2012) or hydrological modelling (e.g., Segoni et al., 2009; Thomas et al., 2018), or obtained from on-site (e.g., Mirus et al., 2018) or remote sensing measurements (e.g., Brocca et al.,

175   2012).

We test whether including antecedent conditions has an informative value on rainfall thresholds by separating rainfall events into 4 subsets, depending on whether they are triggering or not and whether they fall above or below the optimised ED power law threshold. Then for each of the 4 sets and each duration, we compute the average antecedent rainfall, and check if this is different for triggering and non-triggering events. This is the opposite of what is normally done by separating events into with

and without antecedent rainfall a priori (e.g., Frattini et al., 2009). In order to ensure we have enough events for this analysis we utilise the longest record available (RDI 1972–2018) and events with duration up to 6 days. Averaging the antecedent rainfall allows us to see general trends, not focusing on every individual event.

## 3   Results

### 3.1   Daily and hourly thresholds

We define several rainfall thresholds by maximising TSS for the different rainfall datasets, as well as the associated timeframes (Figure 2). Comparing the results for the three different rainfall products (comparison D in Figure 2 or top panel for TSS values in Figure 3) it can be seen as expected that performance is best with the high quality hourly rainfall product which uses high resolution radar information for the disaggregation of daily sums (RHIR). Disaggregation using the closest hourly rain gauge




(RHIG) seems to lead to worse performances than the corresponding daily analysis RDI (red and blue bars in the upper part of
Figure 3). However this may be deceptive, as the time periods as well as the number of landslides behind the rainfall datasets
are different. This is a critical point we investigate below.

A fairer comparison would be to compare performances over the same time period (05/2003–12/2010) and considering the
same landslide events (comparison A in Figure 2 and middle panel of Figure 3). In this case, the differences in performances
across the different rainfall datasets become smaller. The hourly disaggregated product using radar (RHIR) is still leading to the
best performance, but the performance with daily data (RDI) is improved even with the simple disaggregation using the closest
rain gauge (RHIG), for all rainfall properties except duration. Remarkably, the daily rainfall dataset RDI retains reasonably
good predictive power despite its coarser temporal resolution.

One additional comparison that can be made in the overlapping timeframe (05/2003–12/2010) is with all landslide events,
regardless of whether the timing is also known or only the date. The performances obtained with daily data and all these events,
are now comparable to the ones with the high quality hourly product (RHIR). The differences between the two are even more
evident looking at the thresholds associated with the performances shown here. The daily thresholds considering only events
with known timing rather than all landslides decrease to 22.5 mm/d for the maximum intensity (32.0 mm/d considering all
landslides within the timeframe), 35.0 mm for the total rainfall (47.9 mm considering all landslides within the timeframe), and
14.0 mm/d for the mean intensity (19.0 mm/d considering all landslides within the timeframe).

While the decrease of the thresholds and performances is consistent for all rainfall properties as the landslide dataset is
reduced, this is not a general result. Rather it demonstrates that the size and accuracy of the landslide dataset is important, and
that results based on shorter records are likely to be less robust as they are more susceptible to individual events, years, outliers,
or mistakenly reported landslides. This short-record bias is also evident when comparing daily thresholds obtained using the
1981–2018 timeframe or the shorter timeframe 05.2003-12.2010 for which RHIR is available (first and third bars in the bottom
panel of Figure 3). The thresholds obtained with the latter are higher. The reason is that in 2005, 187 landslides occurred, most
of them due to a single intense summer storm in August. Considering all 38 years (1981–2018) the effect of that "outlier" year
is reduced as it amounts to ca. 10% of the total number of landslides available with known timing (almost 40% within the
period 05.2003-12.2010).

Final visual evidence of the lower robustness of thresholds defined using hourly rainfall data, is found in the relative fre-
quency plots of triggering events for hourly rainfall data, compared to daily (upper and lower portions of Figure 4). The trig-
gering events at the hourly resolutions (634 events) are much more sparse than the corresponding daily events (2117 events).

### 3.2 Inaccurate landslides timing: triggering and peak intensities

Results of two different approaches are presented here to illustrate the case when historical landslide inventories have no timing
information available. The landslides are assigned to the actual timing of the database, the most intense hour within the actual
day, or the most intense hour within a 48h window centred around the actual timing.

We defined ED threshold using each of these modified landslide datasets (Figure 2). Searching for the most intense hour
within the actual day of the landslides (# 6 in Figure 2) leads to optimal thresholds that are not far off the ones defined





using the actual timing (#3 in Figure 2). Instead, when the hour with the maximum intensity is found within a 48h window centred on the actual timing (#7 in Figure 2), the threshold changes, leading to a higher coefficient $a$ and smaller slope $b$.

This observation is true for both threshold optimisation using TSS or following the frequentist approach, for which the change in the threshold parameters is present also when limiting the time to the day of the landslides. The explanation for this difference is that the TSS maximisation approach for the definition of ED thresholds is relatively robust, as altering the timing of the landslides some triggering events might change their total rainfall and duration values, but non-triggering events are unaffected. What is important is that for the TSS maximisation in both scenarios of unknown adjusted timing, the TSS value

associated with the best threshold is higher than if the timing was known.

All the observations presented here are valid also when carrying out the same analysis over the 1981–2018 time period using RHIG. The TSS maximisation leads to basically identical thresholds in the 3 scenarios but the TSS increases from 0.65 (actual timing) to 0.67 (most intense hour within the actual date), or 0.70 (most intense hour within a 48h window). Following the frequentist approach, the 5% exceedance the TSS also increases from 0.44 (actual timing) to 0.51 (most intense hour within

the actual date), or 0.60 (most intense hour within a 48h window).

This means that if we do not know the timing of landslides accurately and assign them to some a priori decided rainfall event property, then we are overestimating the landslide prediction skill of our ED curves. Extending this to a situation in which the actual timing is unknown and this technique is applied to compensate for it, while the threshold might not be very far off, the user would overestimate model performance leading to a false overconfidence in his/her predictions.

Nevertheless, having to make a choice between the two methods of correcting timing, limiting the search of the rainiest hour to the actual date, seems to be slightly better, with smaller overestimation of the performances (TSS), and threshold curve parameters more similar to the ones obtained using the actual timing. Considering a 48 h window not only leads to overestimation of the TSS, but the thresholds are also affected. For both threshold definition methods, the threshold in this case gets higher (higher $a$) and less steep (smaller $b$).

### 245    3.3   Rainfall quality: gauge density and interpolation

To test the importance of the general quality of the rainfall dataset in the context of the daily-hourly temporal resolution comparison, we use here the hourly gauge measurements (RHG) in a sparse network and the hourly gridded rainfall dataset (RHIG). The latter, takes advantage of the high quality daily record (RDI), which is based on a denser daily rain gauges network and accounts for climatology and topography (Comparison C in Figure 2).

As before, the comparison between the different rainfall datasets should not be based on the thresholds obtained, since both triggering and non-triggering events can potentially change, but rather on the landslide prediction performances associated with them. When the rain gauge rainfall record is used directly (RHG), whether duplicated at each (closest) landslide location (#5 in Figure 2) or just using one timeseries per gauge (#4 in Figure 2), the sensitivity drops, and so does the TSS. The two rainfall datasets (RHIG and RHG) have exactly identical hourly rainfall fractions and differ only by the daily sum, which

for RHIG is forced to match the RDI daily rainfall of the corresponding cell. When using the station hourly timeseries, the triggering rainfall events have generally smaller event characteristics than the corresponding RHIG events. Out of total 634




events, 423 have smaller maximum intensity, 382 have smaller mean intensity, 447 have smaller total rainfall, and 461 have shorter duration. This results in a decrease of the maximum TSS of up to 0.07, mostly due to a lower sensitivity (for total rain, the sensitivity drops from 0.72 to 0.63).

The same drop in performance is observed when following the frequentist approach (Figure 2). The TSS, which is 0.44 for the analysis using the hourly timeseries adjusted with the daily product (RHIG), drops to 0.29 or 0.24, depending on whether the susceptible cells or rain gauge locations are used. In this case the effect on the threshold (ED curve) is also very consistent: the curves are lower (smaller $a$) and slightly steeper (higher $b$). This is a consequence of the fact that it is especially the short (intense) events that are missed (underestimated) when considering rainfall measurements further away from the actual location

of landslides (RHG rather than RDI).

### 3.4   Rainfall normalisation

The improvement achieved by defining thresholds not based directly on the values of the different precipitation characteristics, but scaling them by a certain quantile of the corresponding event characteristic, a certain quantile of daily/hourly precipitation, or the mean annual precipitation is shown in Figure 5 for the daily RDI and hourly RHIR datasets. When searching for the event

property thresholds, it seems to be irrelevant which quantile is chosen, as the TSS seems to be only slightly fluctuating around a value somewhere between the no normalisation and the mean annual precipitation lines. Completely different behaviour is observed for the normalisation using quantiles of hourly/daily rainfall. In that case, performances comparable to the other cases are achieved only for the highest quantiles, especially for hourly data (right panels in Figure 5).

    In general, best performances are obtained with normalisation by mean annual precipitation. In fact, with hourly data, this

level of performance can only be reached for few very high rainfall quantiles of the total rainfall (centre right in Figure 5). With daily data instead, performances are comparable with the mean daily precipitation and a wider range of quantiles ($q > 0.4$) of daily rainfall and event properties.

    The results for the RDN normalisation (not shown here) are basically indistinguishable from the MAP, not in terms of value of optimum threshold, but of performances, with differences in the TSS of the normalised optimum threshold of less than 0.01.

The improvement of landslide prediction with normalised rainfall thresholds is statistically demonstrated, but it demands a physical explanation. We hypothesise that the reason lies in the fact that the rainfall regime (climate) and the landsliding process (erosion) are connected through the landscape balance between weathering and soil formation, and the rainfall-driven erosion of the top soil by landsliding and other processes (e.g. Norton et al., 2014). In climates with a highly erosive rainfall regime and high topography, the rate of landsliding has adjusted to match the lower soil formation rates. Consequently, we

need on the average higher rainfall intensities to generate landslides there. The scaling of rainfall thresholds by a high-intensity rainfall quantile corrects for landscape scale differences between these process rates and leads to better prediction of landslide occurrence regionally. Evidence for this hypothesis can be found in some studies (e.g., Leonarduzzi et al., 2017; Peruccacci et al., 2017) and can also be observed by comparing the differences in triggering intensities to those of mean daily precipitation values in our data (Figure 6). Here cells in which the mean daily precipitation is higher, also have generally higher triggering





intensities. Accounting for this in the threshold definition, for example dividing the values of maximum intensity to the MAP of the corresponding cell, results in an improvement in the performances.

It is interesting to note that most of the rainfall triggering intensities are indeed among the strongest intensities recorded. Most of the triggering intensities (circles in Figure 6) lie between the 0.75 and 1 quantiles of rainfall. This is the foundation for the success of rainfall thresholds for landslide prediction.

**3.5    Antecedent rainfall**

Including antecedent wetness or rainfall on regional/national scale thresholds is not a simple task. In fact, while antecedent rainfall - triggering rainfall thresholds are successful in many local studies (e.g. for the Seattle area,  Chleborad, 2003), the results shown in Section 3.4 are indicative of the heterogeneity at the regional/national scale which will make antecedent rainfall signals difficult to detect. For example, the approach suggested in Chleborad (2003) of defining thresholds considering

as variables the 3-day and the 15-day prior cumulative rainfall applied to the RDI data 1972–2018 shows no pattern useful for the definition of thresholds. Nevertheless, the information content even in the simplest proxy of soil wetness, that is the antecedent rainfall, is clear (Bogaard and Greco, 2018).

In our experiment where we separated the events into observed triggering or non-triggering, and predicted triggering or not-triggering (above or below the ED threshold obtained maximising TSS) and plotting the mean antecedent rainfall for 5 and

30-day periods, we can see that antecedent rainfall can explain some of the misclassifications generated by the ED threshold (Figure 7). We expect that some of the misses (triggering events below ED curve) were actually landslides caused by low rainfall amounts on very wet soil. At the same time some false alarms (non-triggering events above ED curve) were wrongly predicted as triggering and weren't due to a very low antecedent rainfall. These are exactly the patterns we observe in Figure 7. Higher intensity events are generally associated with higher antecedent rainfall, due to seasonality effects (typically in the

wetter periods of the year), the false alarms are associated with clearly smaller antecedent rainfall than the true positives, and, even more importantly, the misses have, for almost all durations, higher antecedent rainfall than the false alarms. As expected, the true negative events have on average the smallest antecedent rainfall for most durations.

The highest antecedent rainfall for misses (triggering events below ED curve) for events of duration of 1 day could be indicative of the importance of antecedent conditions, either because the wrong event has been identified as triggering or

because those are really triggered due to previous high soil wetness conditions rather than the event itself. However, we cannot provide evidence that this is the case. The patterns for the 5 and 30 days antecedent rainfall look very similar, showing that the antecedent conditions are consistent over longer periods. The only difference is in the true negatives, which for the 30 days, have a much smaller mean antecedent rainfall than the other events.





## 4 Discussion

In the work presented here we show that the choice of the optimal temporal resolution for the definition of rainfall thresholds might not be a straightforward exercise, and that many more aspects should be taken into consideration before concluding that the highest temporal resolution is best for landslide prediction.

We argue that from a theoretical point of view, hourly rainfall data are superior to daily data as they can capture the short convective events lasting few hours which are known to trigger landslides and which get averaged out in the daily sum. This

has also been confirmed in some studies, e.g. Marra (2019) and Gariano et al. (2019), where different temporal resolutions were compared to show the underestimation of thresholds at lower temporal resolution. Also in the work presented here, when we consider the exact same time period and landslide events, we see that performances at the hourly temporal resolution are superior to those at the daily resolution, especially for high quality datasets (RHIR). On the other hand, we show with this work that there are several additional factors that should be taken into consideration.

Choosing hourly rainfall data usually implies dealing with shorter historical records, lower quality (sparser) rainfall datasets, and less rich landslide databases. Typically in the past rain gauges were mostly recording precipitation daily, which means that the daily datasets go further back in time, allowing for an analysis spanning over many more years. Taking the example of Switzerland, since 1961 ca. 420 gauges are available for generating the RDI rainfall product. The first hourly gauges start to appear around 1981, and only 45 of those are consistently measuring until 2018. The much lower density of hourly rain

gauges makes the quality of the interpolated product lower, or the distance between observed landslide and measured rainfall locations greater, and therefore less representative. In recent years (ca. since 2012) the number of hourly gauges has increased dramatically, with 270 stations at the moment, but this would allow an analysis on maximum 7 years (compared to the 48 years available at the daily resolution). The variability in the optimum threshold for the different time periods is proof of the risk of using shorter timeframes (lower panel in Figure 3).

At the hourly resolution also the richness of the landslide database is affected, as not only the date but also the timing of the landslide must be known. Staley et al. (2013) addressed this issue and showed the overestimation of thresholds when considering peak rainstorm instead of triggering intensity, when the actual timing is unknown. This generally leads to overestimation of the maximum intensity, but potentially also other event parameters. Here, especially when the threshold is obtained maximising TSS, the optimum threshold does not seem to change much, at least if the landslide date is known. This seem to be a better

choice whenever possible, as allowing a larger window (48h cantered on the actual timing) leads to a bigger threshold change, both if maximising TSS or following the frequentist approach. Nevertheless, in both cases, the performances are overestimated if the peak intensity is used to time the landslide, giving the user overconfidence in the threshold values themselves.

Some last factors to take into consideration when choosing the temporal resolution, are that in many countries hourly records of rainfall could be even shorter and of lower quality than in Switzerland, and choosing to work with daily data might be even

more important. Furthermore, thinking of utilising rainfall thresholds in an operational setting, daily forecasts are usually more reliable than hourly forecasts (e.g. Shrestha et al., 2013).





In all the comparisons between hourly and daily rainfall, we purposely refrained from comparing the value of the optimal thresholds and of the ED curves between hourly and daily analysis. In fact, to allow this comparison, strong assumptions must be made, which are clearly not realistic, such as assuming that the daily intensity is 24 times the corresponding hourly intensity.

This is in agreement with the recommendation in Gariano et al. (2019) and other studies to not extend daily ED or ID rainfall thresholds into the sub-daily domain.

Two methods for rainfall threshold estimation were presented here, to show that the threshold optimisation method used does not impact the main conclusions. While our work does not intend to compare the two methods, the results presented here show clearly that accounting for triggering events also in the definition of the threshold (e.g. maximising TSS) increases the

robustness of the obtained threshold. In fact, while the performances and the parameters of the ED curves are affected in both cases, the frequentist approach seems to be more sensitive, with greater differences in optimal thresholds and greater variability in performance (e.g. see variability of the optimum ED thresholds in Figure 2). Nevertheless, there might be conditions in which rainfall records are not available and only triggering events can be reconstructed from newspaper and other historical records. In those conditions, a method like the frequentist approach would be the only option.

Lastly, we make an argument in our work for the benefits of normalising the rainfall thresholds. The different normalisation methods we test show that high quantiles of rainfall intensities, quantiles of event properties, MAP and RDN are all valuable parameters to be used, especially with daily rainfall data. Nevertheless, we suggest using MAP as a general and widely available climatological variable.

## 5 Conclusions

We define and test rainfall thresholds for triggering of landslides by taking advantage of a rich landslide database and several rainfall products available in Switzerland with the main objective of providing a realistic comparison between hourly and daily rainfall resolutions. We explore the impacts of other issues, like shorter datasets, unknown timing, and more sparse networks, that usually accompany higher temporal resolution data, and we test the impacts of two typical analysis steps in threshold definition: normalisation of the threshold, and antecedent rainfall.

Our main findings are:

- Although hourly rainfall is more appropriate for landslide prediction, several aspects should be taken into consideration before utilising it exclusively for threshold definition. Generally, hourly rainfall records are shorter (only available in recent years), and of lower quality (e.g. based on sparser rain gauge networks), landslide database only seldom contain accurate timing.

- In ideal conditions, hourly datasets do show best predictive performance for landslides, but daily data are not far behind. The benefits of hourly resolutions can be masked by the higher uncertainties in threshold estimation connected to using short records and unknown timing.



- Whenever continuous rainfall records are available together with a landslide inventory, we recommend including also non-triggering events in the definition of the optimal thresholds, not only because false alarms are an essential factor in warning systems, but also to increase the robustness of the threshold estimation.

- Localisation of rainfall thresholds through normalisation is a useful procedure, which allows to compensate for the spatial heterogeneity in rainfall regimes and landslide erosion process rates. We recommend using mean annual precipitation or a high quantile of rainfall intensity as a normalisation factor as an alternative to regionalisation.

- Antecedent rainfall as a proxy of soil wetness state can explain some of the false alarms in rainfall thresholds, associated with lower antecedent rainfall, and some of the misses, preceded by heavy rainfall, even when considering an entire (heterogeneous) country. Although we did not formulate new rainfall-duration curves including antecedent rainfall, it is likely that these will increase predictive skill.

*Data availability.* The rainfall products were provided by the Swiss Federal Office of Meteorology and Climatology MeteoSwiss (available for research purposes upon request). The gauges timeseries are available upon request at https://gate.meteoswiss.ch/idaweb/ (last accessed 18.11.2019). The Swiss Federal Research Institute WSL provided the landslide data (available for research purposes upon request).

*Author contributions.* E. L. conducted the analysis and interpreted the results. P. M. and E. L. conceived the research and prepared the paper.

*Competing interests.* The authors declare that they have no conflict of interest.

*Acknowledgements.* This research was funded by the Swiss National Science Foundation grant 165979 awarded to P. M.





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

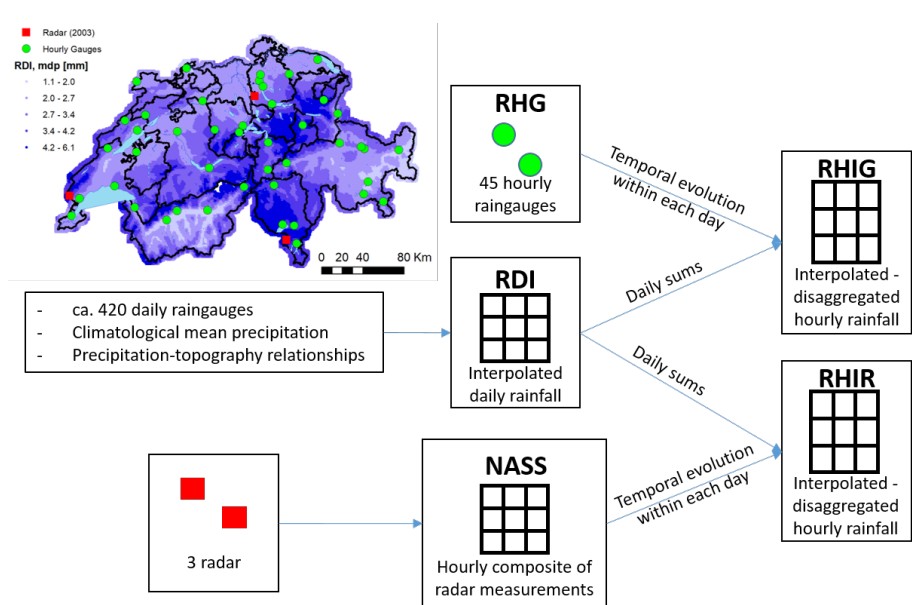

**Figure 1.** Map and scheme of the different rainfall datasets used in the analysis. The daily interpolated product (RDI), the hourly rain gauges (RHG), and the two derived hourly gridded products, which preserve the daily sums from RDI, but use the sub-daily temporal variability of a radar composite (RHIR) or of the hourly rain gauges (RHIG).





**Table 1.** Description of the different rainfall datasets used.

| | Rainfall Daily Interpolated | Rainfall Hourly Gauges | Rainfall Hourly Interpolated Gauges | Rainfall Hourly Interpolated Radar |
|---|---|---|---|---|
| acronym | RDI | RHG | RHIG | RHIR |
| data source | rain gauges | rain gauges | rain gauges | rain gauges + radar |
| type of product | gridded ($1km^2$) | gauges (Figure 1) | gridded ($1km^2$) | gridded ($1km^2$) |
| temporal resolution | daily | hourly | hourly | hourly |
| timeframe | 1972–2018 | 1981–2018 | 1981–2018 | 05.2003-12.2010 |
| methods | interpolation of rain gauges using climatology and topography relationships | measured | disaggregation of RDI using temporal evolution of RHG | disaggregation of RDI using temporal evolution of radar data |
| reference | Frei and Schär (1998) | - | - | Wüest et al. (2010) |
| number of landslides in timeframe | 2271 | 1842 (634 with known timing) | 1842 (634 with known timing) | 501 (237 with known timing) |
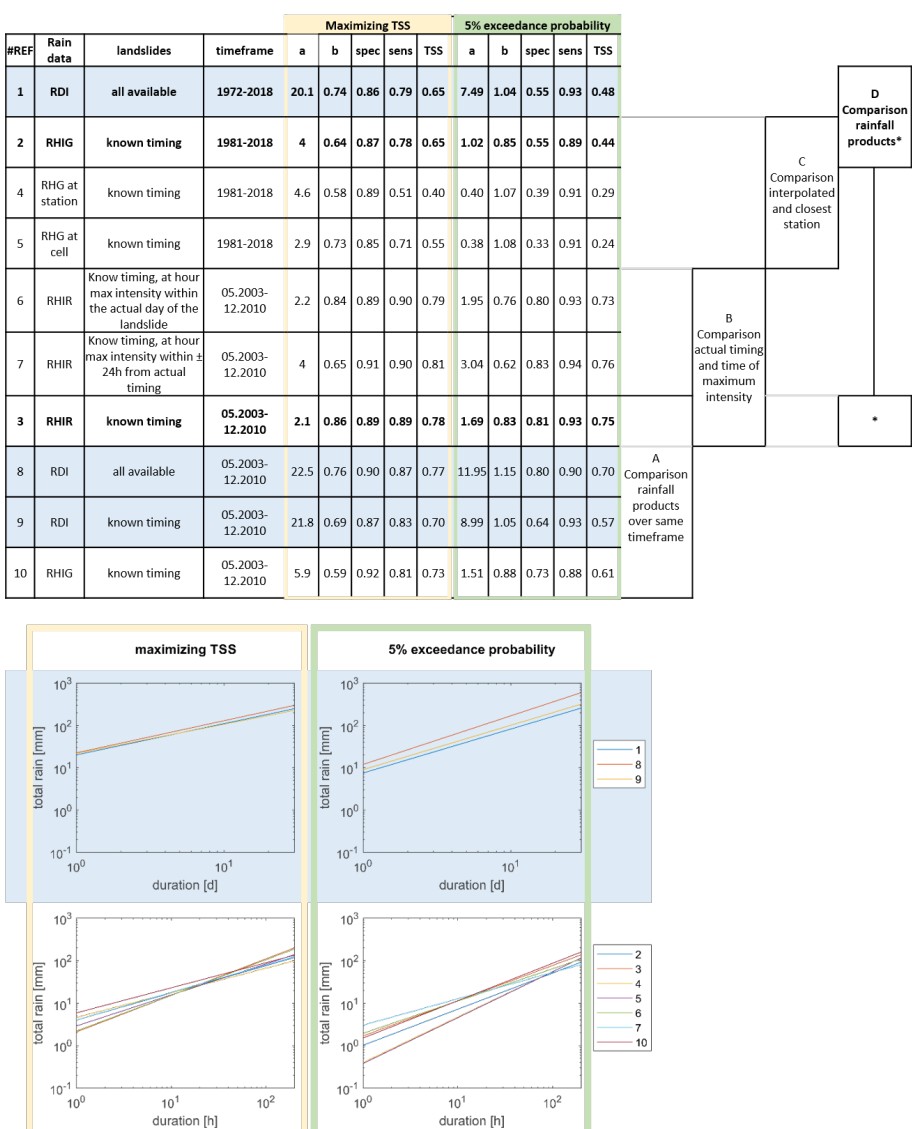

**Figure 2.** Table containing the coefficient of the threshold power law curve in the total rainfall - duration plane obtained by maximising the TSS, or selecting the 5% exceedance probability line following the frequentist approach, for all the different timeframes and rainfall records. To facilitate reading, the different comparisons carried out are indicated, matching the respective results. On the right, all the ED threshold curve are shown, separated into daily (above) and hourly (below), obtained with TSS maximisation (left) or following the frequentist approach (right). The numbers in the legend match the "#REF" entry in the Table above.


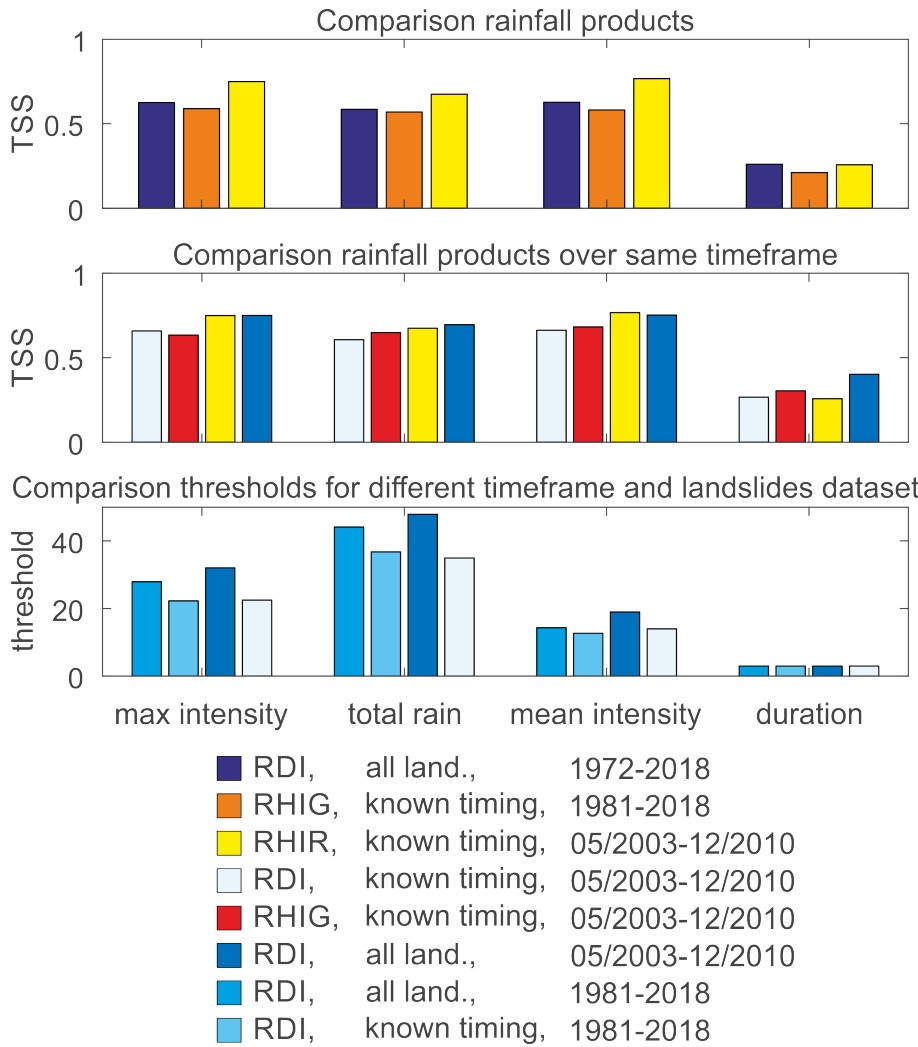

**Figure 3.** True skill statistic for the different precipitation characteristics and all the different rainfall dataset considered. Above using for each rainfall dataset the entire timeframe available, below by comparing over the overlapping timeframe (05.2003-12.2010).

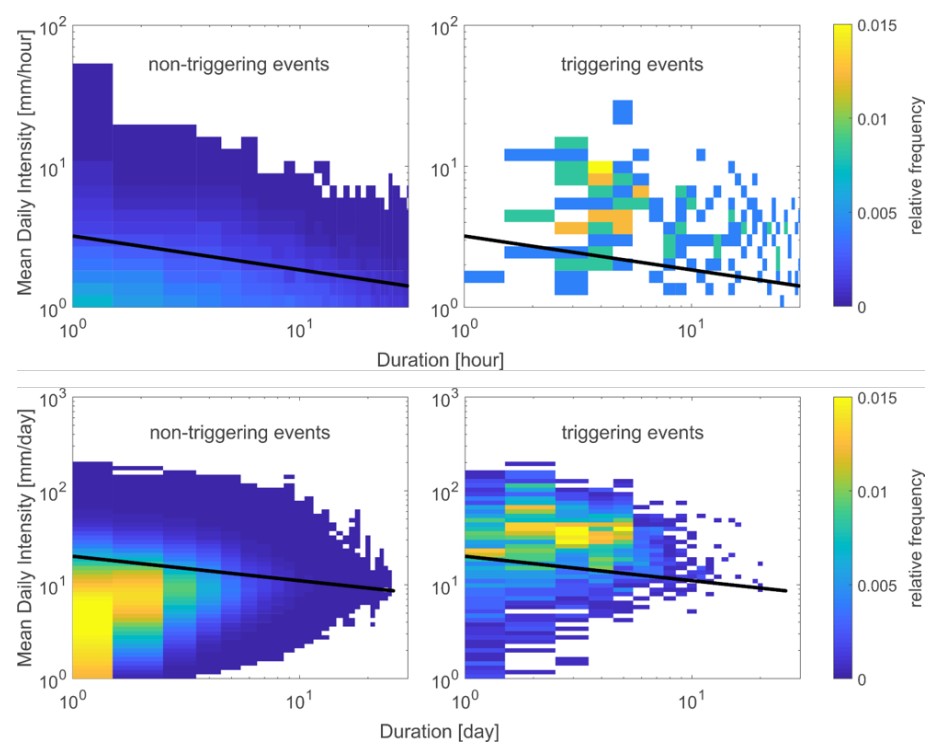

**Figure 4.** Total rainfall - duration (ED) plots with color scale representing the relative frequency of non triggering (left) and triggering (right) events. The lines represent the best power law curve thresholds obtained maximising TSS, above with hourly (RHIR) and below with daily (RDI) rainfall data.

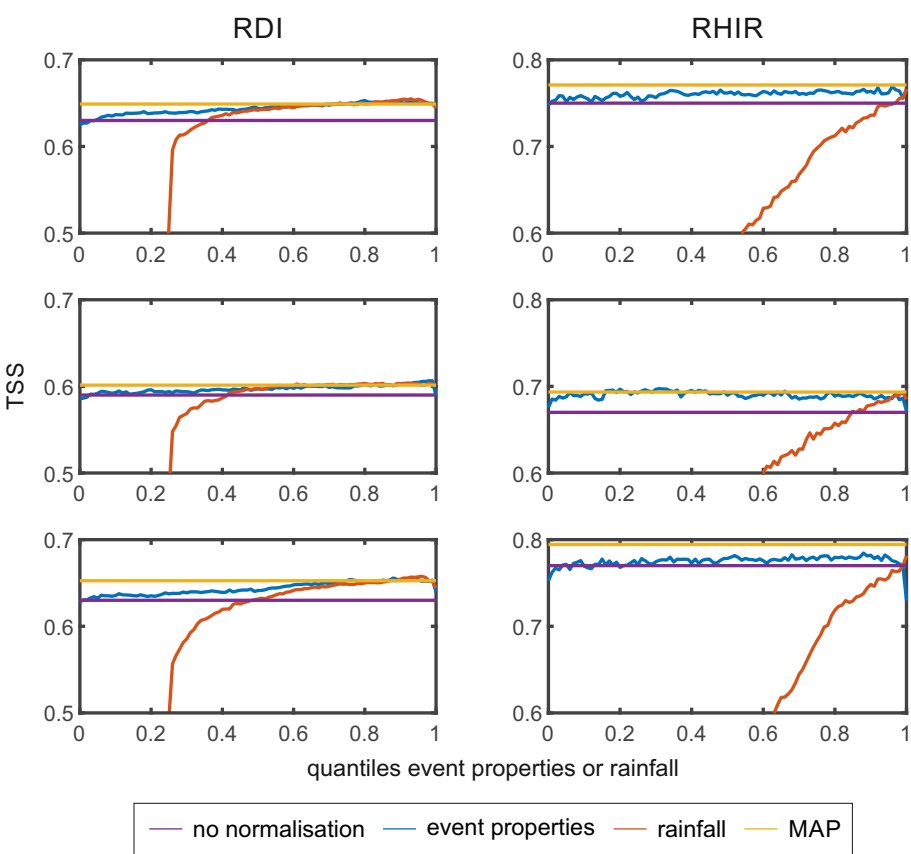

**Figure 5.** True Skill Statistic values for the best threshold for the different normalisations, for the daily (RDI, left) and hourly (RHIR, right) rainfall data. On top for maximum rainfall, in the middle for total rain, and the bottom for mean intensity. For the normalisation by event properties (event properties) and quantiles of rainfall (rainfall), the TSS is computed for each 0.01 quantile value (x axis). For the normalisation by mean annual precipitation (MAP) and the TSS value of the variable without normalisation (no normalisation), the constant value of the TSS is indicated as a straight line across all x values.

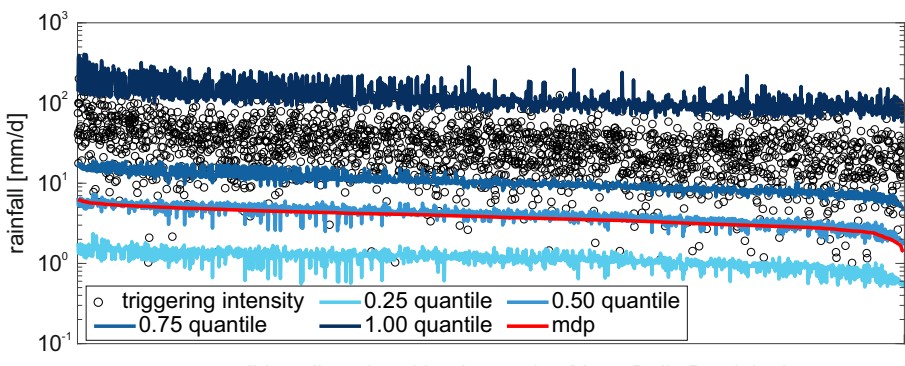

**Figure 6.** Quantiles of daily intensities, Mean Daily Precipitation (MAP/365) and maximum daily triggering intensities for all the susceptible cells (rainfall cells with at least one landslide), sorted by value of mean daily precipitation (cell with the highest to the lowest MDP, left to right in the x axis). Markers show the daily rainfall intensities of triggering events for each cell.

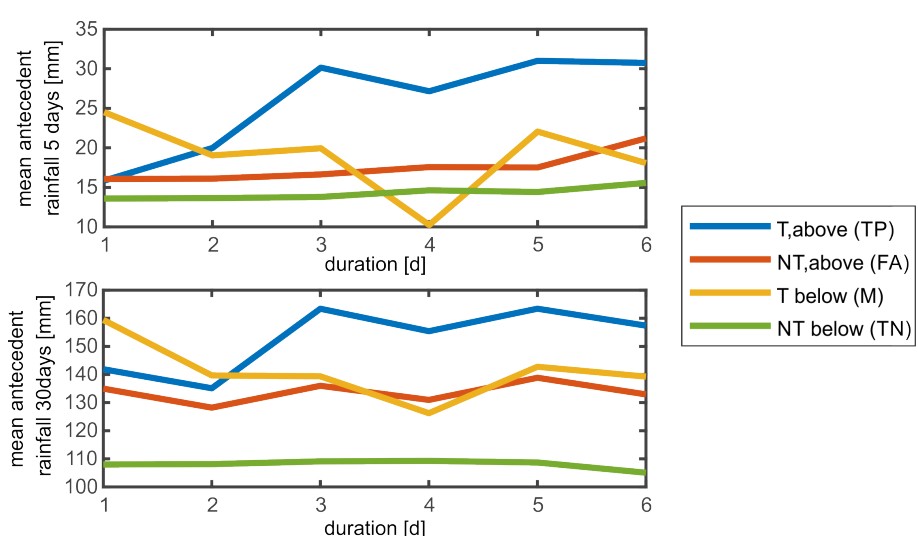

**Figure 7.** Mean antecedent rainfall over 5 (above) or 30 days before the date beginning of the event. All the rainfall events are separated into: True Positives (observed triggering events which are above the ED threshold), False Alarms (observed non-triggering events which are above the ED threshold), Misses (observed triggering events which are below the ED threshold), and True Negatives (observed non-triggering which are below the ED thresholds), and the mean antecedent rainfall is computed for each of these for each event duration (1-6 days). Results are based on the RDI rainfall dataset, 1972-2018.