# Peer review of "Deriving rainfall thresholds for landsliding at the regional scale: daily and hourly resolutions, normalisation, and antecedent rainfall"

_Natural Hazards and Earth System Sciences, 2020_

## Referee Comment (RC1) · Francesco Marra (Referee) · 22 May 2020

The study examines the performance of rainfall thresholds for landslides obtained from different hourly and daily datasets, as well as the use of normalizations for the threshold localization and some preliminary analyses on the impact of antecedent conditions.

The manuscript is well written, the study well fits the topics of this journal and is carried out with sound methods and data. To my view, the novelty brought by the study is that it collects from an amount of recent theoretical and smaller-scale studies and collectively examines the practical implications using a large dataset on a wide alpine region. I believe it contributes to our practical knowledge on rainfall thresholds for landslides

triggering and therefore deserves publication. Overall, it was a pleasant reading.

I list below a few comments for the author's consideration.

Kind regards,

Francesco Marra

1. In Section 3.4, did you check the results using absolute quantiles (corresponding to return levels, or probabilities in time) instead to wet quantiles? To my view, the number of wet days contributes generating the local climatology (indeed it does for return levels) and the wet-quantiles somehow forget this. I refer in particular to the discussion in lines 281-291, which I believe would hold more for absolute quantiles. Also, it would be very interesting to see fig 6 with absolute quantiles (return levels) instead of wet-quantiles. I'm not saying this must be included, rather that it should be checked before exclusion (even though I'd be personally interested in seeing the figure anyways)

2. I suggest including ID/ED thresholds in the results in Fig. 3. Many readers are familiar with such thresholds and it would be helpful for the quantitative interpretation of the results

3. Do your archives contain information on the landslides type? Are debris flows included in the database? I would guess that debris flows, generally triggered by short convective events, are more subject to the temporal resolution. If relevant, is there a way to check this from your data? Also, in the discussion (lines 355-356) you recommend "not to extend daily thresholds . . . into the sub-daily domain" – can this recommendation be made more explicit from the elements in your hands?

4. The title focuses on the temporal resolution aspect while the paper provides quite a lot of additional information. Perhaps you can consider expanding it

5. Lines49-54: More details on the point (b) (i.e. poor matching of landslide in space) should be provided in the introduction as this is a crucial to the findings. There are few lines afterwards but I think some (more) lines are needed in the introduction as well

6. Line 58: what do "(analysis steps)" refer to?

7. Line 97: could you provide more details on the optimization (what was optimized, how, why)?

8. Line 181-182: "Averaging..." this sentence was not clear to me. Also what do you mean by "trends"?

9. Line 195: it looks like RDI retains good predictive power because of the stations density, is this correct? Are there other reasons?

10. Line 215: perhaps I did not understand: why does the sparseness of the points in the figure imply lower robustness? It this not just a consequence of the data sample?

11. Line 323-329: I'd argue that Marra 2019 do not claim/confirm that higher resolutions are superior as no evaluation of the predictive performance was done. Rather, systematic differences are highlighted, with consequences for the physical interpretation of the triggering amounts and the quantitative comparison of thresholds and threshold parameters obtained from different datasets

12. Fig. 2 and 3 took me some time to understand. I could not find suggestions on how to make them more immediately understandable, but I feel it is something to communicate within the review

---

## Referee Comment (RC2) · Ben Mirus (Referee) · 23 May 2020

This NHESS Discussions paper provides a detailed and objective investigation of numerous factors related to development of rainfall thresholds for landslide forecasting. It relies on a database of landslide occurrence across Switzerland and four alternative configurations of rainfall data in daily and hourly resolutions. Beyond the issue of data temporal resolution, the authors investigate the effect of uncertainty in landslide timing, sparseness of rain gage data, duration of records, normalization of rainfall thresholds for different regions, and the role of antecedent rainfall in threshold performance.

Overall it is a very relevant topic and a very nice contribution. In fact, it provides quite a

few surprising and constructive insights that can inform future considerations for landslide threshold development, so I wonder if the title could be rephrased to reflect the various novel contributions of the work, not just the limitations? Ultimately, the paper should definitely be published in NHESS with some quite minor revisions. In particular, the investigation of antecedent conditions was not entirely clear to me, so the description of the methods and results could be improved. Otherwise, numerous edits would enhance the clarity of other aspects of the study, which I have outlined by line number below.

1: When it comes to landslides, I have started to prefer "forecast" over "predict" since it implies less specificity on location and/or timing. Also, for a concise abstract one could delete phrases such as "In this paper" as it's not needed.

2: You are not quite providing a comprehensive evaluation of "landslide prediction performance," since that can take many forms, but rather specifically of "rainfall threshold performance."

15: Avoid ending on a negative note. Perhaps rephrase to state that is it worth the additional effort to build antecedent rainfall into threshold curves?

21: In a new paper we provide further updates and review of reports on economic losses in the U.S. as well as analysis of over 300,000 landslides (Mirus et al., Landslides, 2020, DOI: 10.1007/s10346-020-01424-4).

50: Specify that you focus on "different temporal resolution of data." Even though this does also relate to the negative consequence of lower density and duration of rainfall measurements.

60-61: Might be worth clarifying that these studies have in fact demonstrated the utility of including antecedent conditions, but at a relatively narrow scale comparted to the effort you explore here. However, as you know, Wicki et al. (Landslides, 2020, DOI 10.1007/s10346-020-01400-y) have already evaluated soil moisture at the regional

scale for landslide warning. Also, probably our other paper from 2018 is more appropriate for citing here related to comparing antecedent rainfall and soil state (Mirus et al., Landslides, 2018, DOI: 10.1007/s10346-018-0995-z).

62: It's not clear what a realistic comparison means, so it might be more accurate to state ". . . an extensive, objective comparison between real rainfall data at hourly and daily resolutions for. . ."

67: What is "TSS"? Should introduce all acronyms before using and also repeat definitions in figure captions and tables for clarity.

80-83: This is a bit unclear and maybe includes several typos or confusing phrasing, so I had to re-read a few times:

- Rainfall not raninfall

- Clarify that you used two different hourly gridded data, not two-hourly gridded rainfall. Just avoid that source of confusion.

- Initially it was unclear how hourly data could be derived from RDI, so I thought it was a typo until later reading the disaggregation methods.

Suggested revision: "We used two different hourly datasets that were derived by disaggregating the RDI such that the daily sums match that of the corresponding RDI cell at the same 1 x 1 km resolution."

87: Is it possible to give a range of distances to explain what you mean by "quite sparse"?

89: It may be unclear to some readers what the fourth record is. You have only described the daily RDI and two hourly records RHIR and RHIG (derived using the RDI and RHG). Consider listing out all four record names here to avoid confusion.

175: Consider adding Thomas et al. (WRR, 2019, DOI: 10.1029/2019WR025577P) here as well regarding investigations into satellite measurements for landslide warning.

179-180: Since this is the opposite of what is normally done, I think a slightly more detailed explanation is needed. I was not able to fully grasp the methods or interpretation of the results in Figure 7.

239: Consider listing "his/her/their", using only the pronoun "their," or more simply revising to "overconfidence in the threshold predictions."

300: As you note, the 3-15 day approach of Chleborad and others is indeed specific to the Seattle area. It would be possible to evaluate what time-scales are most appropriate for distinguishing between rainfall linked to the "trigger" versus the "cause" as outlined by Bogaard and Greco (2018). We re-evaluated the appropriate timescales for ID and cumulative rainfall thresholds (Scheevel, Baum, Mirus, and Smith, 2017; doi: 10.3133/ofr20171039) as well as rainfall-saturation thresholds (Mirus, Morphew, and Smith, 2018, already cited herein). Thus, it is possible that other timescales are better for Switzerland. Can you clarify which timescales you tested, how, and why those times were selected? Again, the methodology for considering antecedent conditions was not totally clear to me.

308: Confusing. Revise to clarify that they were wrongly predicted as triggering, but no known landslides occurred due to low antecedent rainfall.

340-348: This paragraph has a number of typos and grammatical errors relative to the rest of the paper, so perhaps these were overlooked in the authors final editing?

343: Careful to clarify that the threshold / triggering intensity is over-estimated, not the measured max. rainfall itself being overestimated.

356: This may be a good point to mention that whereas hourly records capture some indication of rainfall intensity, daily rainfall totals tend to represent a metric somewhere in between the intensity of the storm and its cumulative depth. It might provide another explanation for why the daily still performs reasonably well, since not all landslides are triggered by brief, high-intensity events.

357: Consider specifying which two methods.

364: The only method? I'm sure that some alternative approaches could be proposed by others. Consider avoiding such an absolute statement.

365: For brevity: "Lastly, we demonstrate the benefits of normalizing the rainfall thresholds using high quantiles of rainfall intensities, quantiles of event properties, MAP, or RDN. These are all particularly useful when using daily data, but we suggest MAP as it is general and a widely available climatological variable."

371: Still not clear what you mean by "realistic comparison." Suggest: ". . . of providing a rigorous and objective comparison between. . ."

372: ". . .unknown landslide timing, and more sparse rain gage networks . . ."

376: Suggest: ". . . more appropriate for forecasting landslides since it better captures triggering intensities, several other aspects. . ."

380: Suggest: ". . . daily data are not far behind, potentially since it does tend to capture cumulative storm totals that may also be relevant for landslide triggering." [?]

383-385: This is true and useful, but unlike your other conclusions, it is nothing new. Consider clarifying that your results further underscore/reinforce previous findings about the importance of non-triggering events.

392: ". . .these would increase. . .."

Figure 1. Define NASS and mdi. Increase legend size for RDI in Swiss map.

Table 1. I think you know the dates of all the landslides during the various time periods, no? If so, please revise "known timing" should really be "known date and time" to avoid any confusion. Same for the subsequent figures.

Figure 2. Define all acronyms used in figure and caption.

Figure 4. It's a bit surprising that the duration in the upper plots (hours) have the same

[Figure]

axis numbers (10ˆ0 and 10ˆ1) as the duration does in the lower plots (days).

Figure 6: Define acronyms like MAP and MDP. I'm not sure I understand the x-axis label and there are no numbers. Please clarify.

Figure 7. Consider labelling (a) and (b) or clarifying that lower plot is Mean Antecedent rainfall (MAR) for 30d. Again, define all acronyms used in caption or legend. These results are a bit confusing and I'm not sure the methods or results are explained clearly enough. What does the "duration" refer to? Duration of the triggering storm event?
* * *

---

## Author Response (AR1)

Prof. Thomas Glade
Editor
NHESS
17.07.2020

Dear Prof. Glade:

We resubmit the manuscript nhess-2020-125 entitled *Deriving rainfall thresholds for landsliding at the regional scale: daily and hourly resolutions, normalisation, and antecedent rainfall*. We thank the Dr. Marra and Dr. Mirus for their thoughtful and constructive comments. We have addressed every reviewer comment below (in italics) but want to draw your attention to the following general areas of revision:

1.      We modified the title to: *Deriving rainfall thresholds for landsliding at the regional scale: daily and hourly resolutions, normalisation, and antecedent rainfall*

2.      We added the results for normalisation using absolute quantiles (in addition to wet quantiles) and modified sections 2.5 and 3.4, and Figure 5 accordingly

3.      We added the results for ED threshold performances in the two upper panels in Figure 3

4.      We extended and improved explanation of the analysis of antecedent rainfall (Section 2.6)

5.      We updated Figure 4, with the correct ED plots and thresholds

All modifications in the manuscript's text are highlighted in yellow in the new version.

We thank you for your efforts in handling this manuscript and look forward to hearing your decision soon.

Thanks and best regards,

Elena Leonarduzzi and Peter Molnar

*The study examines the performance of rainfall thresholds for landslides obtained from different hourly and daily datasets, as well as the use of normalizations for the threshold localization and some preliminary analyses on the impact of antecedent conditions. The manuscript is well written, the study well fits the topics of this journal and is carried out with sound methods and data. To my view, the novelty brought by the study is that it collects from an amount of recent theoretical and smaller-scale studies and collectively examines the practical implications using a large dataset on a wide alpine region. I believe it contributes to our practical knowledge on rainfall thresholds for landslides triggering and therefore deserves publication. Overall, it was a pleasant reading.*

*I list below a few comments for the author's consideration.*

*Kind regards,*

*Francesco Marra*

*1. In Section 3.4, did you check the results using absolute quantiles (corresponding to return levels, or probabilities in time) instead to wet quantiles? To my view, the number of wet days contributes generating the local climatology (indeed it does for return levels) and the wet-quantiles somehow forget this. I refer in particular to the discussion in lines 281-291, which I believe would hold more for absolute quantiles. Also, it would be very interesting to see fig 6 with absolute quantiles (return levels) instead of wet-quantiles. I'm not saying this must be included, rather that it should be checked before exclusion (even though I'd be personally interested in seeing the figure anyways)*

It is true that return periods and rainfall probabilities should consider both intermittency (frequency of rain) and rain intensity, so the question raised is an important one. We show the results for the normalisation using both "wet" and "absolute" quantiles in Figure 1 here below. Please note that to simplify the plot here, we removed the event properties line, which is present in the original Figure 5 in the manuscript. The "wet quantiles" line matches the "rainfall" line in the original manuscript's Figure 5. We tried to limit the number of normalisation factors considered for simplicity and clarity, and chose wet quantiles because only high absolute quantiles were usable (i.e. > 0 for all susceptible cells, from 0.65 quantile for daily rainfall and 0.9 for hourly rainfall).

As can be seen from Figure 1, using absolute quantiles doesn't seem to differ much from the wet quantiles when using daily rainfall (left side of Review-Figure 1) for the highest quantiles (much worse for lower quantiles). When using hourly rainfall, absolute quantiles do seem to outperform wet quantiles and, in the case of total rainfall, also the normalization using mean annual precipitation. Nevertheless, because the normalization with the MAP seems overall quite robust, and MAP is generally an easily available climatological variable, we still would recommend using MAP as a normalization parameter. We added the results for normalisation using absolute quantiles (in addition to wet quantiles) and modified sections 2.5 and 3.4, and Figure 5 accordingly.

2. I suggest including ID/ED thresholds in the results in Fig. 3. Many readers are familiar with such thresholds and it would be helpful for the quantitative interpretation of the results

We agree that ID/ED are the type of threshold readers are more familiar with, nevertheless the idea of Figure 3 is to be complementary to Figure 2. The optimal ED curve parameters as well as the corresponding TSS are reported in Figure 2 for all the different dataset considered (same cases as in

Figure 3, and more). We believe that adding also the ED results in Figure 3 would make it less readable as it would require two extra entries (a and b parameter of the curve in the bottom part of the figure) and a different y axis. We have updated the figure adding the TSS values for the ED thresholds in the first 2 panels of Figure 3.

3. Do your archives contain information on the landslides type? Are debris flows included in the database? I would guess that debris flows, generally triggered by short convective events, are more subject to the temporal resolution. If relevant, is there a way to check this from your data? Also, in the discussion (lines 355-356) you recommend "not to extend daily thresholds … into the sub-daily domain" – can this recommendation be made more explicit from the elements in your hands?

The Swiss flood and landslide damage database which we used here contains flood, debris flow, landslide and rockfall events. We used only those classified as "landslides", therefore debris flow shouldn't be included (although the classification is based on the "primary process", so debris flow might have occurred also as secondary process).

The recommendation of not extending daily thresholds to the subdaily domain we report is actually a conclusion from Gariano et al. (2019), as mentioned in line 355, and support our belief that hourly thresholds should not be derived (i.e. extrapolated) from daily resolution. Furthermore, thresholds obtained with daily or hourly data cannot be compared since strong and unrealistic assumptions must be made to allow the comparison (as mentioned in the manuscript in the lines just prior). A weak confirmation of this is that the optimum thresholds when considering the 2003-2005 timeframe and only landslides with known date and timing for mean intensity are 1.34mm/h and 14mm/d (definitely not a 24 factor). Both aspects are not really a conclusion drawn from any analysis carried out here, but rather a general recommendation we believed was important.

4. The title focuses on the temporal resolution aspect while the paper provides quite a lot of additional information. Perhaps you can consider expanding it

We have changed the title to: "Deriving rainfall thresholds for landsliding at the regional scale: daily and hourly resolutions, normalisation, and antecedent rainfall"

5. Lines49-54: More details on the point (b) (i.e. poor matching of landslide in space) should be provided in the introduction as this is a crucial to the findings. There are few lines afterwards but I think some (more) lines are needed in the introduction as well

In the introduction, we list 3 different undesired consequences of choosing higher temporal resolution rainfall in the context of landslide forecasting with rainfall thresholds. We later expand on the issue of rain gauge density in section 3.3.

6. Line 58: what do "(analysis steps)" refer to?

It refers to the two aspects mentioned right after (normalization and use of antecedent wetness). We removed "(analysis steps)" to avoid confusion.

7. Line 97: could you provide more details on the optimization (what was optimized, how, why)?

We modified the sentence to: "We choose 24 hours for daily rainfall data and 6 hours for hourly rainfall data. The hourly interstorm period of 6 hours separating events is selected as the one leading to the

best performances (highest True Skill Statistic, see methodology explained hereafter), within a range of 2-12 hours, which is the amount of dry hours expected to separate individual storms. This is longer than the requirement of statistical independence between events, which Gaál et al. (2014) showed to be at least 2 hours."

difference reflects the role that antecedent rain plays in landslide generation.

*8. Line 181-182: "Averaging…" this sentence was not clear to me. Also what do you mean by "trends"?*

The idea is that because we're considering an entire (heterogeneous) country, there are differences which don't allow to see on an event scale the signal in the antecedent wetness (which is the reason for example why the MAP normalization improves the results). As an example of this, following the approach developed in Chleborad 2003 for the Seattle area using triggering and antecedent rainfall (see also Review-Figure 2), the data across the whole of Switzerland don't show any sort of clustering or threshold behaviour as it does on the local scale (mentioned in lines 299-301 of the manuscript).

We expanded and improved the explanation of the antecedent rainfall analysis accordingly (lines 181-193).

*9. Line 195: it looks like RDI retains good predictive power because of the stations density, is this correct? Are there other reasons?*

In general, the good performances of RDI are due to its high quality (station density, interpolation method), which definitely play a role also here. But in this case we're also suggesting that the signal at the daily scale might be stronger than expected. In fact, we're removing the advantages of the daily resolution we mention here (longer record, more landslide events), but still the difference in performances compared to hourly data is not dramatic. Nevertheless, we cannot quantify this effect exactly, and separate how much of the performances at the daily scale are due to the higher station density or better interpolation (higher data quality).

*10. Line 215: perhaps I did not understand: why does the sparseness of the points in the figure imply lower robustness? It this not just a consequence of the data sample?*

The sparseness is definitely a consequence of the sample size. It just visually shows how, when the size is so small, a few "wrong" points could affect the results and change the thresholds. That the robustness depends on the sample size (especially with uncertain data) is a pretty obvious concept, but we thought visualizing the effect could help the reader grasp the issue and its consequences.

*11. Line 323-329: I'd argue that Marra 2019 do not claim/confirm that higher resolutions are superior as no evaluation of the predictive performance was done. Rather, systematic differences are highlighted, with consequences for the physical interpretation of the triggering amounts and the quantitative comparison of thresholds and threshold parameters obtained from different datasets*

Changed to: "Previous studies (e.g., Marra, 2019; Gariano et al., 2020) have focused on the effect of temporal resolution, and showed that using lower temporal resolutions leads to the underestimation of the thresholds. From a theoretical point of view, we argue that hourly rainfall data are superior to daily data as they can capture the short convective events lasting few hours which are known to trigger landslides and which get averaged out in the daily sum. Also in the work presented here, when we consider the exact same time period and landslide events, we see that performances at the hourly

temporal resolution are superior to those at the daily resolution, especially for high quality datasets (RHIR). On the other hand, we show with this work that there are several additional factors that should be taken into consideration."

*12. Fig. 2 and 3 took me some time to understand. I could not find suggestions on how to make them more immediately understandable, but I feel it is something to communicate within the review*

We added in Figure 3 a reference to the corresponding comparison in Figure 2, and the performances of ED thresholds. These should allow to better link the two figures.

[Figure]

*Review-Figure 1 . True Skill Statistic values for the best threshold for the different normalisations, for the daily (RDI, left) and hourly (RHIR, right) rainfall data. On top for maximum rainfall, in the middle for total rain, and the bottom for mean intensity. For the normalisation by wet and absolute quantiles of rainfall, the TSS is computed for each 0.01 quantile value (x axis). For the normalisation by mean annual precipitation (map) and the TSS value of the variable without normalisation (no normalisation), the constant value of the TSS is indicated as a straight line across all x values.*

[Figure]

[Figure]

Figure 2a. Preliminary graph showing estimates of 3-day and prior 15-day cumulative precipitation associated with historical landslides that were part of events with 3 or more landslides in a 3-day period, in Seattle (filled triangles). The solid red line is a lower-bound threshold (visually identified) for the initiation of landslides when the 15-day cumulative is 3.0 inches or less. The dashed horizontal line is a lower-bound threshold, that was tentatively proposed, for conditions of 15-day antecedent precipitation exceeding 3 inches. (Chleborad, 2000).

*Review-Figure 2. Left, the cumulative 3 days rainfall up to landslide days and the corresponding prior 15-days cumulative rainfall, following the approach in Chleborad (2003). Right, the original figure from Chleborad (2003) for Seattle Area. Rainfall amounts are reported in inches to facilitate comparison.*

*This NHESS Discussions paper provides a detailed and objective investigation of numerous factors related to development of rainfall thresholds for landslide forecasting. It relies on a database of landslide occurrence across Switzerland and four alternative configurations of rainfall data in daily and hourly resolutions. Beyond the issue of data temporal resolution, the authors investigate the effect of uncertainty in landslide timing, sparseness of rain gage data, duration of records, normalization of rainfall thresholds for different regions, and the role of antecedent rainfall in threshold performance.*

*Overall it is a very relevant topic and a very nice contribution. In fact, it provides quite a few surprising and constructive insights that can inform future considerations for landslide threshold development, so I wonder if the title could be rephrased to reflect the various novel contributions of the work, not just the limitations? Ultimately, the paper should definitely be published in NHESS with some quite minor revisions. In particular, the investigation of antecedent conditions was not entirely clear to me, so the description of the methods and results could be improved. Otherwise, numerous edits would enhance the clarity of other aspects of the study, which I have outlined by line number below.*

*1: When it comes to landslides, I have started to prefer "forecast" over "predict" since it implies less specificity on location and/or timing. Also, for a concise abstract one could delete phrases such as "In this paper" as it's not needed.*

We have implemented the suggested changes

*2: You are not quite providing a comprehensive evaluation of "landslide prediction performance," since that can take many forms, but rather specifically of "rainfall threshold performance."*

We have changed the sentence as suggested

*15: Avoid ending on a negative note. Perhaps rephrase to state that is it worth the additional effort to build antecedent rainfall into threshold curves?*

Rephrased to: "Finally, while antecedent rainfall thresholds approaches used at the local scale are not successful at the regional scale, we demonstrate that there is predictive skill in antecedent rain as a proxy of soil wetness state, despite the large heterogeneity of the study domain."

*21: In a new paper we provide further updates and review of reports on economic losses in the U.S. as well as analysis of over 300,000 landslides (Mirus et al., Landslides, 2020, DOI: 10.1007/s10346-020-01424-4).*

We added the suggested reference

*50: Specify that you focus on "different temporal resolution of data." Even though this does also relate to the negative consequence of lower density and duration of rainfall measurements.*

We changed the sentence as suggested

*60-61: Might be worth clarifying that these studies have in fact demonstrated the utility of including antecedent conditions, but at a relatively narrow scale comparted to the effort you explore here. However, as you know, Wicki et al. (Landslides, 2020, DOI 10.1007/s10346-020-01400-y) have already evaluated soil moisture at the regional scale for landslide warning. Also, probably our other paper from 2018 is more appropriate for citing here related to comparing antecedent rainfall and soil state (Mirus et al., Landslides, 2018, DOI: 10.1007/s10346-018-0995-z).*

We were planning on citing Wicki et al. in the revised manuscript, unfortunately it wasn't yet published when we first submitted. We also added a reference to Mirus et al., 2018. A sentence has been added mentioning the scale of previous studies as suggested: "… and the inclusion of antecedent rainfall which provides additional information on soil state prior to landsliding, typically studied at local scales (Glade et al., 2000; Godt et al., 2006; Mirus et al., 2018a, b)". We also added the citation of Wicki et al. later in the text.

*62: It's not clear what a realistic comparison means, so it might be more accurate to state "… an extensive, objective comparison between real rainfall data at hourly and daily resolutions for…"*

Rephrased to: "(a) to provide an extensive comparison between hourly and daily rainfall data for the definition of rainfall thresholds, considering several practical consequences of choosing a higher temporal resolution,"

*67: What is "TSS"? Should introduce all acronyms before using and also repeat definitions in figure captions and tables for clarity.*

We have added explanations of acronyms

*80-83: This is a bit unclear and maybe includes several typos or confusing phrasing, so I had to re-read a few times:*
*- Rainfall not raninfall*
We corrected the typo
*- Clarify that you used two different hourly gridded data, not two-hourly gridded rainfall. Just avoid that source of confusion.*

We changed the sentence as suggested

*- Initially it was unclear how hourly data could be derived from RDI, so I thought it was a typo until later reading the disaggregation methods.*
*Suggested revision: "We used two different hourly datasets that were derived by disaggregating the RDI such that the daily sums match that of the corresponding RDI cell at the same 1 x 1 km resolution."*

We changed the sentence as suggested

*87: Is it possible to give a range of distances to explain what you mean by "quite sparse"?*

It is visible in Figure 1. We added a reference to the figure, as well as an estimate of the average density (ca. 1 rain-gauge per 900 km$^2$)

*89: It may be unclear to some readers what the fourth record is. You have only described the daily RDI and two hourly records RHIR and RHIG (derived using the RDI and RHG). Consider listing out all four record names here to avoid confusion.*

We added the list of rainfall product names as suggeted

*175: Consider adding Thomas et al. (WRR, 2019, DOI: 10.1029/2019WR025577P) here as well regarding investigations into satellite measurements for landslide warning.*

We added the suggested reference

*179-180: Since this is the opposite of what is normally done, I think a slightly more detailed explanation is needed. I was not able to fully grasp the methods or interpretation of the results in Figure 7.*

We always work with rainfall events as defined at the beginning of section 2.2. Therefore, by "duration" we refer to the actual duration of the events defined accordingly (given the number of dry hours to separate individual events). All of these events were observed as triggering (if a landslide happened during or right after them) or non-triggering otherwise. According to the optimum ED threshold, we can separate them also into predicted triggering (above the curve) or non-triggering. The intersections of these prediction/observation gives us the 4 groups: false alarms, true predictions, misses, and true negatives. If the antecedent rainfall is the parameter explaining the "ED failures", we would expect that misses were associated with high antecedent rainfall, and false alarms with very low antecedent rainfall. We investigate this by averaging within each of the 4 groups the antecedent rainfall for each event duration. We decided to do it separately for each event duration (all events of duration 1 day, all events of duration 2 days etc.), because we suspected there could be differences also relative to the duration (which could also be a proxy of storm type).
We expanded the description in the paper accordingly, to better introduce and explain the methodology.

*239: Consider listing "his/her/their", using only the pronoun "their," or more simply revising to "overconfidence in the threshold predictions."*

We fixed this

*300: As you note, the 3-15 day approach of Chleborad and others is indeed specific to the Seattle area. It would be possible to evaluate what time-scales are most appropriate for distinguishing between rainfall linked to the "trigger" versus the "cause" as outlined by Bogaard and Greco (2018). We re-evaluated the*

*appropriate timescales for ID and cumulative rainfall thresholds (Scheevel, Baum, Mirus, and Smith, 2017; doi: 10.3133/ofr20171039) as well as rainfall-saturation thresholds (Mirus, Morphew, and Smith, 2018, already cited herein). Thus, it is possible that other timescales are better for Switzerland. Can you clarify which timescales you tested, how, and why those times were selected? Again, the methodology for considering antecedent conditions was not totally clear to me.*

We didn't focus on the triggering vs antecedent rainfall threshold. We simply followed the exact approach of the Seattle area, supported by the fact that the optimum (TSS maximization) threshold for event duration was of 3 days. We decided not to try to improve and optimize this, but rather chose to follow a different approach to verify if indeed the antecedent wetness signal was visible even over such a scale. Hopefully the explanation in response to the comment of lines 179-180 improves the understanding of the approach and methodology.

*308: Confusing. Revise to clarify that they were wrongly predicted as triggering, but no known landslides occurred due to low antecedent rainfall.*

Changed to: "At the same time some false alarms (non-triggering events above ED curve) were wrongly predicted as triggering, but no landslide was observed due to the very low antecedent rainfall"

*340-348: This paragraph has a number of typos and grammatical errors relative to the rest of the paper, so perhaps these were overlooked in the authors final editing?*

We revised the paragraph as follows: "At the hourly resolution also the richness of the landslide database is affected, as not only the date but also the timing of the landslide must be known. Staley et al. (2013) addressed this issue and showed the overestimation of thresholds when considering peak rainstorm instead of triggering intensity. This is common practice, when the actual timing of the landslides is unknown. It generally leads to overestimation of the triggering events' maximum intensity, but potentially also other triggering events' parameters. Here, the optimum threshold does not seem to change much, especially when the threshold is obtained maximising TSS. This is true if at least the landslide date is known. Constraining the timing of landslides to the actual date seems a better choice whenever possible. Allowing a larger window (48h centred on the actual timing) leads to bigger threshold changes, both if maximising TSS or following the frequentist approach. Nevertheless, in both cases, the performances are overestimated if the peak intensity is used to time the landslides, giving the user overconfidence in the threshold values themselves."

*343: Careful to clarify that the threshold / triggering intensity is over-estimated, not the measured max. rainfall itself being overestimated.*

Changed to: "It generally leads to overestimation of the triggering events' maximum intensity, but potentially also other triggering events' parameters."

*356: This may be a good point to mention that whereas hourly records capture some indication of rainfall intensity, daily rainfall totals tend to represent a metric somewhere in between the intensity of the storm and its cumulative depth. It might provide another explanation for why the daily still performs reasonably well, since not all landslides are triggered by brief, high-intensity events.*

We agree with your reasoning. In fact, we believe that hourly intensity can be considered a real intensity, close to the physical process, while daily rainfall is more representative of a weather system

(rather than a storm). This is also one of the reasons why in some climates with longer storms triggering landslides, daily thresholds work. We added a sentence about this aspect in the second conclusion (line 402)

*357: Consider specifying which two methods.*

We specified the methods in the text

*364: The only method? I'm sure that some alternative approaches could be proposed by others. Consider avoiding such an absolute statement.*

We're definitely not suggesting the frequentist approach is the only method. By saying "a method like the frequentist approach would be the only option" we mean any methodology that only considers triggering events, such as the frequentist. We rephrased to: "In those conditions, a method based only on triggering events would be the only option.".

*365: For brevity: "Lastly, we demonstrate the benefits of normalizing the rainfall thresholds using high quantiles of rainfall intensities, quantiles of event properties, MAP, or RDN. These are all particularly useful when using daily data, but we suggest MAP as it is general and a widely available climatological variable."*

We shortened this paragraph.

*371: Still not clear what you mean by "realistic comparison." Suggest: "… of providing a rigorous and objective comparison between…"*

By realistic we mean that it not only considers hourly or daily rainfall, but also takes into account the implications of choosing hourly rainfall (i.e. shorter rainfall records, lower quality rainfall records, less landslide events available) mentioned several times in the manuscript. Rephrased to: "providing a comparison between hourly and daily rainfall resolutions, which considers data limitations associated with choosing a higher temporal resolution"

*372: "…unknown landslide timing, and more sparse rain gage networks…"*

Changed as suggested

*376: Suggest: "…more appropriate for forecasting landslides since it better captures triggering intensities, several other aspects…"*

Changed as suggested

*380: Suggest: "…daily data are not far behind, potentially since it does tend to capture cumulative storm totals that may also be relevant for landslide triggering." [?]*

As mentioned in the comment relative to lines 356, we agree that rainfall intensities and events at the daily and hourly temporal resolution have different meaning. As mentioned above, we believe daily rainfall is rather representative of weather systems and cumulative storm properties which last in most cases less than 1 day. We added: "potentially since daily data tend to capture cumulative storm totals that may also be relevant for landslide triggering"

*383-385: This is true and useful, but unlike your other conclusions, it is nothing new. Consider clarifying that your results further underscore/reinforce previous findings about the importance of non-triggering events.*

It is definitely not a new conclusion, but due to the large number of studies in which this is still not done (without explicitly mentioning why it was not possible to include non-triggering) we still believe it's an important message. We will improve the conclusion accordingly: "Whenever continuous rainfall records are available together with a landslide inventory, our work underscores the importance of including non-triggering events in the definition of optimal rainfall thresholds, not only because false alarms are an essential factor in warning systems, but also to increase the robustness of the threshold estimation."

*392: "…these would increase…"*

Changed as suggested

*Figure 1. Define NASS and mdi. Increase legend size for RDI in Swiss map.*

We increased the size of the legend, removed "NASS" from the figure, since it's a not needed acronym, and the explanation of mdp.

*Table 1. I think you know the dates of all the landslides during the various time periods, no? If so, please revise "known timing" should really be "known date and time" to avoid any confusion. Same for the subsequent figures.*

We revised as suggested

*Figure 2. Define all acronyms used in figure and caption.*

We added all missing acronyms explanations

*Figure 4. It's a bit surprising that the duration in the upper plots (hours) have the same axis numbers (10ˆ0 and 10ˆ1) as the duration does in the lower plots (days).*

Thanks for this comment. We realized that the figure in the manuscript is an older version where ID thresholds (instead of ED) are used and the axis for the hourly data is indeed cut. We updated the Figure.

*Figure 6: Define acronyms like MAP and MDP. I'm not sure I understand the x-axis label and there are no numbers. Please clarify.*

The Mean Daily Precipitation (MDP) as the name says is simply the average daily precipitation (equal to Mean Annual Precipitation / 365). We chose to use the MDP because it's quantitatively comparable to the other precipitation estimates reported in Figure 6, while the MAP would require a secondary axis and be more difficult to directly compare. We explained acronyms and rephrased the caption to improve the explanation.

*Figure 7. Consider labelling (a) and (b) or clarifying that lower plot is Mean Antecedent rainfall (MAR) for 30d. Again, define all acronyms used in caption or legend. These results are a bit confusing and I'm not sure the methods or results are explained clearly enough. What does the "duration" refer to? Duration of the triggering storm event?*

Yes, as mentioned here above, "duration" always refers to events' duration (triggering or non), with events defined given the interarrival time (as described in section 2.2). We added the definition of T (triggering) and NT (non-triggering), and improved the caption explanation.

[Figure]

*Review-Figure 3 Total rainfall - duration (ED) plots with color scale representing the relative frequency of non triggering (left) and triggering (right) events. The lines represent the best power law curve thresholds obtained maximising True Skill Statistic, above with hourly (RHIR) and below with daily (RDI) rainfall data.*